META-RESEARCH

# The changing career paths of PhDs and postdocs trained at EMBL

**Abstract** Individuals with PhDs and postdoctoral experience in the life sciences can pursue a variety of career paths. Many PhD students and postdocs aspire to a permanent research position at a university or research institute, but competition for such positions has increased. Here, we report a time-resolved analysis of the career paths of 2284 researchers who completed a PhD or a postdoc at the European Molecular Biology Laboratory (EMBL) between 1997 and 2020. The most prevalent career outcome was Academia: Principal Investigator (636/2284=27.8% of alumni), followed by Academia: Other (16.8%), Science-related Non-research (15.3%), Industry Research (14.5%), Academia: Postdoc (10.7%) and Non-science-related (4%); we were unable to determine the career path of the remaining 10.9% of alumni. While positions in Academia (Principal Investigator, Postdoc and Other) remained the most common destination for more recent alumni, entry into Science-related Non-research, Industry Research and Non-science-related positions has increased over time, and entry into Academia: Principal Investigator positions has decreased. Our analysis also reveals information on a number of factors – including publication records – that correlate with the career paths followed by researchers.

**JUNYAN LU‡, BRITTA VELTEN†§, BERND KLAUS†#, MAURICIO RAMM¶, WOLFGANG HUBER, RACHEL COULTHARD-GRAF***

**\*For correspondence:**
rachel.coulthard@embl.de

†These authors contributed equally to this work

**Present address:** ‡Medical Faculty, Heidelberg University, Heidelberg, Germany; §Centre for Organismal Studies and Interdisciplinary Center for Scientific Computing, University of Heidelberg, Heidelberg, Germany; #Häfele SE & Co KG, Nagold, Germany; ¶Institute of Biomedicine, University of Turku, Turku, Finland

**Competing interest:** The authors declare that no competing interests exist.

## Introduction

Career paths in the life sciences have changed dramatically in recent decades, partly because the number of early-career researchers seeking permanent research positions has continued to significantly exceed the number of positions available (*Cyranoski et al., 2011*; *Schillebeeckx et al., 2013*). Other changes have included efforts to improve research culture, growing concerns about mental health (*Evans et al., 2018*; *Levecque et al., 2017*), increased collaboration (*Vermeulen et al., 2013*), an increased proportion of project-based funding (*Lepori et al., 2007*; *Jonkers and Zacharewicz, 2016*) and greater awareness of careers outside academic research (*Hayter and Parker, 2019*). Nevertheless, many PhD students and postdocs remain keen to pursue careers in research and, if possible, secure a permanent position as a Principal Investigator (PI) at a university or research institute (*Fuhrmann et al., 2011*; *Gibbs et al., 2015*; *Lambert et al., 2020*; *Roach and Sauermann, 2017*; *Sauermann and Roach, 2012*).

Data on career paths in the life sciences have become increasingly available in recent years (*Blank et al., 2017*; *Council for Doctoral Education, 2020*), and such data are useful to individuals as they plan their careers, and also to funding agencies and institutions as they plan for the future. In this article we report the results of a time-resolved analysis of the career paths of 2284 researchers who completed a PhD or postdoc at the European Molecular Biology Laboratory (EMBL) between 1997 and 2020. This period included major global events, such as financial crisis of 2007 and 2008 (*Izsak et al., 2013*; *Pellens et al., 2018*), and also major events within the life sciences (such as the budget of the US National Institutes of Health doubling between 1998 and 2003 and then plateauing; *Wadman, 2012*; *Zerhouni, 2006*).

## Results

EMBL is an intergovernmental organisation with six sites in Europe, and its missions include

scientific training, basic research in the life sciences, and the development and provision of a range of scientific services. The organization currently employs more than 1110 scientists, including over 200 PhD students, 240 postdoctoral fellows, and 80 PIs. EMBL has a long history of training PhD students and postdocs, and the EMBL International PhD Programme – one of the first structured PhD programmes in Europe – has a completion rate of 92%, with students taking an average of 3.95 years to submit their thesis (data for 2015–2019). More recently, EMBL has launched dedicated fellowship programmes with structured training curricula for postdocs.

Data collection for this study was initially carried out in 2017 and updated in 2021. Using manual Google searches, we located publicly available information identifying the current role of 89% (2035/2284) of the sample (*Table 1*). These alumni were predominantly based in the European Union (60%, 1224/2035), other European countries including UK and Switzerland (20%), and the US (11%). For 71% of alumni (1626/2284), we were able to reconstruct a detailed career path based on online CVs and biographies (see Methods). EMBL alumni also ended up in a range of careers, which were classified as follows: Academia: Principal Investigator; Academia: Postdoc; Academia: Other research/teaching/service role; Industry Research; Science-related Non-research; and Non-science-related. We also collected data on different types of jobs within the last three of these career areas (*Table 2*).

On average, the alumni in our sample published an average of 4.5 research articles about their work at EMBL, and were the first author on an average of 1.6 of those articles (Table S1 in *Supplementary file 1*). Overall, 90% of the sample (2047/2284) authored at least one article about their EMBL work, and 73% (1666/2284) were the first author on at least one article. The average time between being awarded a PhD and taking up a first role in a specific career area ranged from 4.2 years for Non-science-related positions to 6.8 years for a Principal Investigator (PI) position.

### Most alumni remain in science

The majority of alumni (1263/2284=55.3%) were found to be working in an academic position in 2021, including 636 who were PIs, 244 who were in Academia: Postdoc positions, and 383 who were working in Academia: Other positions, which included teaching, research and working for a core facility/technology platform (*Figure 1A*). Just under one-sixth (332/2284=14.5%) were employed in Industry Research positons, and a similar proportion (349/2284=15.3%) were employed in Science-related Non-research positions, such as technology transfer, science administration and education, and corporate roles at life sciences companies. Around 4% were employed in professions not related to science, and the current careers of around 11% of alumni were unknown.

**Table 1.** Career outcomes for 2284 EMBL alumni.

| Career | PhD alumni | Postdoc | Total |
|---|---|---|---|
| Academia: PI (AcPI) | 215 (22.2%) | 421 (32%) | 636 (27.8%) |
| Academia: Other (AcOt) | 102 (10.5%) | 281 (21.4%) | 383 (16.8%) |
| Academia: Postdoc (AcPD) | 168 (17.3%) | 76 (5.8%) | 244 (10.7%) |
| Industry research (IndR) | 153 (15.8%) | 179 (13.6%) | 332 (14.5%) |
| Science-related Non-research (SciR) | 178 (18.4%) | 171 (13%) | 349 (15.3%) |
| Non-science-related (NonSci) | 47 (4.9%) | 44 (3.3%) | 91 (4%) |
| Unknown | 106 (10.9%) | 143 (10.9%) | 249 (10.9%) |
| TOTAL | 969 | 1315 | 2284 |

See *Table 2* for more information on the different jobs covered by Industry Research, Science-related Non-research, and Non-science-related. This classification is based on *Stayart et al., 2020*.

AcPI: includes those leading an academic research team with financial and scientific independence – evidenced by a job title such as group leader, professor, associate professor or tenure-track assistant professor. Where the status was unclear from the job title, we classified an alumnus as a Principal Investigator (PI) if one of the following criteria was fulfilled: (a) they appear to directly supervise students/postdocs (based on hierarchy shown on website); (b) they have published a last author publication from their current position; (c) their group website or CV indicates that they have a grant (not just a personal merit fellowship) as a principal investigator. AcOt: differs from *Stayart et al., 2020* in that it includes academic research, scientific services or teaching staff (e.g., research staff, teaching faculty and staff, technical directors, research infrastructure engineers).

Of those who became PIs, 75.3% moved from a postdoc to their first PI position, with 20.6% moving from an Academia: Other position (*Figure 1—figure supplement 1A*). On average, PhD alumni became PIs 6.1 calendar years after their PhD defence, and postdoc alumni became PIs 2.5 years after completing their EMBL postdoc. Almost half of the postdoc alumni who became PIs did so directly after completing their EMBL postdoc (168 of 343). Other postdoc alumni made the transition later, most frequently after one additional postdoc (71 alumni) or a single Academia: Other position (56 alumni). 40 alumni held multiple academic positions between their EMBL postdoc and their first PI position, and eight had one or more non-academic positions during this period.

The career paths of those in other positions were more varied (*Figure 1—figure supplement 1B–E*). For example, for alumni who moved into Industry Research, 20.2% entered their first industry role directly from their PhD, 56.4% from a postdoc position, and 13.3% from Academia: Other positions. Moreover, 71.6% remained in this type of role long-term.

The wide variation in job titles used outside academia makes it difficult to assess career progression, but almost 60% (453/766) of alumni working outside academia had a current job title that included a term indicative of a management-level role (such as manager, leader, senior, head, principal, director, president or chief). For leavers from the last five years (2016–2020), this number was 45% (78/174), suggesting that a large proportion of the alumni who leave academia enter – or are quickly promoted to – managerial positions.

For further analysis, EMBL alumni were split into three 8 year cohorts. More recent cohorts were larger, reflecting the growth of the organization between 1997 and 2020, and also contained a higher percentage of female researchers (*Table 3*). When comparing cohorts, we observed some differences in the specific jobs being done by alumni outside academia 2021 (Table S2 in *Supplementary file 1*). For example, the percentage of alumni involved in 'data science, analytics, software engineering' roles increased from 2% (11/625) for the 1997–2004 cohort to 4% (37/896) for the 2013–2020 cohort. However, the absolute number of alumni for most jobs outside academia was small, so our time-resolved analysis therefore focussed on the broader career areas described above.

## Percentage of EMBL alumni who become PIs is similar to that for other institutions

For all timepoints, the percentages of alumni from the 2005–2012 and 2013–2020 cohorts working in PI positions in 2021 were lower than the percentage for the 1997–2004 cohort (*Figure 1—figure supplement 2*). To assess whether this pattern was specific to EMBL, we compared our data with data from other institutions, noting that different institutions can use different methods to collect data and classify career outcomes. We also note that career outcomes are influenced by the broader scientific ecosystem and by the subject focus of institutions and departments, which may attract early-career researchers with dissimilar career motivations. Nevertheless, comparing long-term outcomes with other institutions allows us to interrogate whether the changes we observe for the most frequent, well-defined and linear career path – the PhD–>Postdoc–>PI career path – reflect a general trend.

A number of institutions have released data on career outcomes for PhD students. Stanford University, for example, has published data on the careers of researchers who received a PhD between 2000 and 2019 (*Stanford Biosciences, 2021*): Stanford has reported that 34% (145/426) of its 2000–2005 PhD alumni were in research-focussed faculty roles in 2018, and that 13% (63/503) of its 2011–2015 PhD alumni were in PI roles; these numbers are comparable to the figures of 37% (78/210) and 11% (25/234) we observe for EMBL alumni for the same time periods (*Figure 1C*). The EMBL data are also comparable to data from the life science division at the University of Toronto (*Reithmeier et al., 2019*; *University Toronto, 2021*): for example, Toronto has reported that 31% (192/629) of its 2000–2003 graduates and 25% (203/816) of its 2004–2007 graduates were in tenure stream roles in 2016; the corresponding figures for EMBL were 39% (52/132) and 28% (49/172).

We also compared our EMBL data with data from the University of Michigan, the University of California at San Francisco, and the University of Chicago, and found similar proportions of alumni entering PI positions for comparable cohorts (*Figure 1—figure supplement 3*). This is consistent with our hypothesis that the differences between cohorts are not EMBL-specific, and reflect a wide-spread change in the number of PhDs and postdocs relative to available PI positions.

**Table 2.** Classification of Industry Research, Science-related Non-research and Non-science-related positions.

| Job function | PhD alumni | Postdoc | Total |
|---|---|---|---|
| **Industry research** | | | |
| R & D scientist* | 138 (14.2%) | 167 (12.7%) | 305 (13.4%) |
| Entrepreneurship† | 6 (0.6%) | 8 (0.6%) | 14 (0.6%) |
| Postdoctoral | 7 (0.7%) | 1 (0.1%) | 8 (0.4%) |
| Business development, consulting & strategic alliances ‡ | 2 (0.2%) | 3 (0.2%) | 5 (0.2%) |
| Total | 153 (15.8%) | 179 (13.6%) | 332 (14.5%) |
| **Science-related Non-research** | | | |
| Administration and training | 35 (3.6%) | 35 (2.7%) | 70 (3.1%) |
| Business development, consulting & strategic alliances | 38 (3.9%) | 20 (1.5%) | 58 (2.5%) |
| Tech support and product development | 20 (2.1%) | 24 (1.8%) | 44 (1.9%) |
| Science writing and communication | 16 (1.7%0) | 21 (1.6%) | 37 (1.6%) |
| Data science, analytics, software engineering § | 15 (1.5%) | 13 (1%) | 28 (1.2%) |
| Intellectual property and law | 16 (1.7%) | 10 (0.8%) | 26 (1.1%) |
| Science education and outreach | 10 (1%) | 11 (0.8%) | 21 (0.9%) |
| Clinical research management | 8 (0.8%) | 4 (0.3%) | 12 (0.5%) |
| Regulatory affairs | 5 (0.5%) | 7 (0.5%) | 12 (0.5%) |
| Clinical services/public health | 4 (0.4%) | 6 (0.5%) | 10 (0.4%0) |
| Sales and Marketing | 4 (0.4%) | 6 (0.5%) | 10 (0.4%) |
| Healthcare provider | 1 (0.1%) | 8 (0.6%) | 9 (0.4%) |
| Other | 4 (0.4%) | 2 (0.2%) | 6 (0.3%) |
| Entrepreneurship | 2 (0.2%) | 2 (0.2%) | 4 (0.2%) |
| Science policy and government affairs | 0 (0%) | 2 (0.2%0) | 2 (0.1%) |
| Total | 178 (18.4%) | 171 (13%) | 349 (15.3%) |
| **Non-science-related** | | | |
| Data science, analytics, software engineering § | 18 (1.9%) | 25 (1.9%) | 43 (1.9%) |
| Business development, consulting & strategic alliances | 16 (1.7%) | 4 (0.3%) | 20 (0.9%) |
| Other (inc retired) | 7 (0.7%) | 12 (0.9%) | 19 (0.8%) |
| Entrepreneurship | 5 (0.5%) | 2 (0.2%) | 7 (0.3%0) |
| Administration and training | 1 (0.1%) | 1 (0.1%) | 2 (0.1%) |
| Total | 47 (4.9%) | 44 (3.3%) | 91 (4%) |

*This function differs from the schema in **Stayart et al., 2020**; it includes alumni carrying out or overseeing scientific research in industry as group leaders, research staff, technical directors and non-directorship research leadership roles, including alumni who appear to be working in computational biology roles of a pharma, biotech, contract research or similar company regardless of job title (i.e. including data science roles that appear to be related to analysis of research-related data.)

†Founders of companies whose primary focus is R&D (including contract research organizations).

‡Includes director-level senior management roles overseeing the scientific direction & research of a company with R&D focus, e.g. CSOs in biotech start-ups.

§Not including computational biology roles linked to R&D functions.

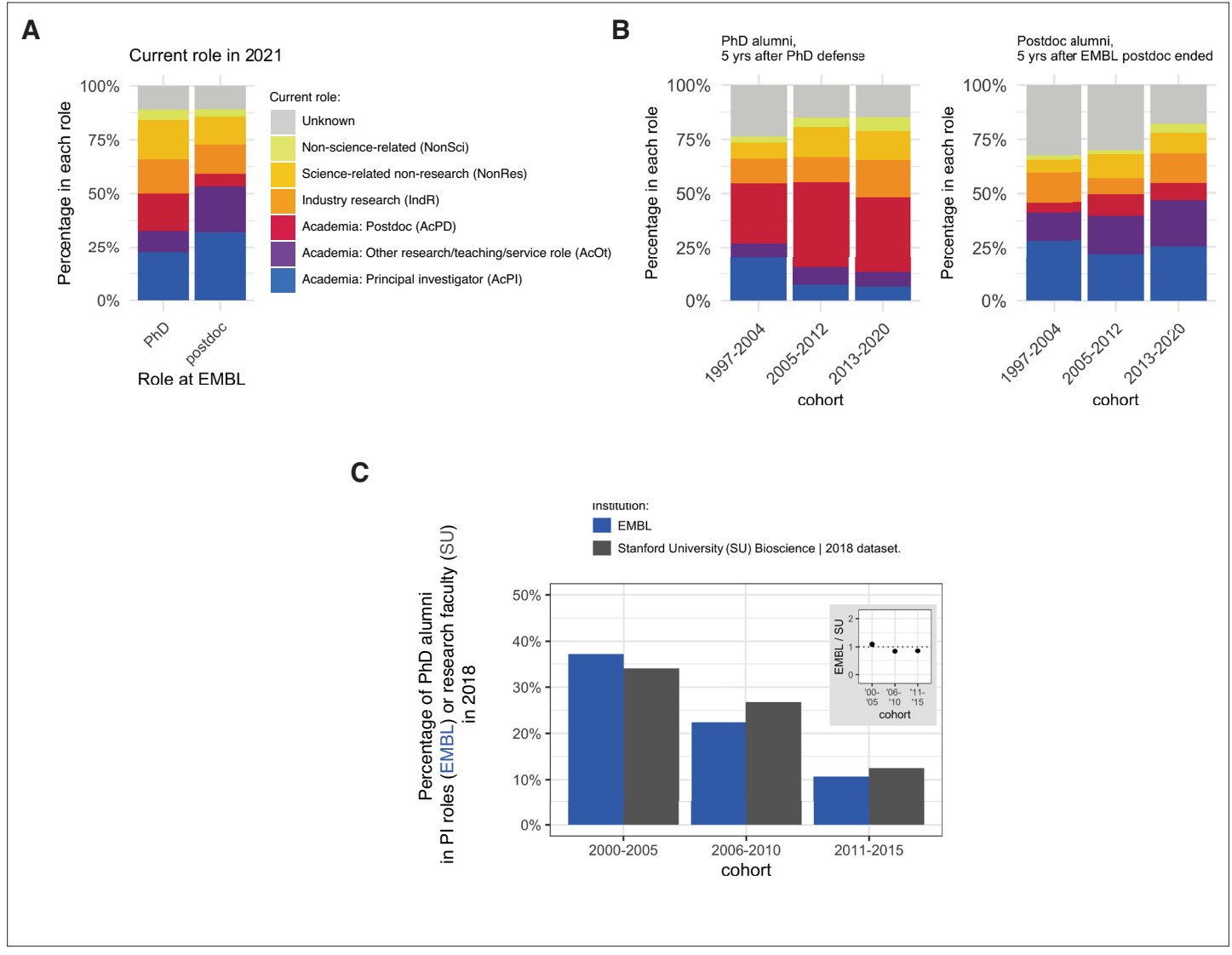

**Figure 1.** Career outcomes for EMBL alumni. (**A**) Charts showing the percentage of PhD alumni (n=969) and postdoc alumni (n=1315) from EMBL in different careers in 2021 (see *Table 1*). (**B**) Charts showing percentage of PhD (left, n=800) and postdoc (right, n=1053) alumni in different careers five years after finishing their PhD or postdoc, for three different cohorts. Chart excludes 169 PhD students and 262 postdocs who have not yet reached the five-year time point. (**C**) Charts showing the percentage of PhD alumni from EMBL (blue column) in PI positions with the percentage of PhD alumni from Stanford University (grey column) in research-focused faculty positions (*Stanford Biosciences, 2021*). Detailed information about the comparison group can be found in Table S3 in *Supplementary file 1*.

The online version of this article includes the following source data and figure supplement(s) for figure 1:

**Source data 1.** Summary data plotted in *Figure 1A, B*.

**Figure supplement 1.** Sankey diagrams showing movement between different careers.

**Figure supplement 1—source data 1.** Summary data plotted in the simplified Sankey plots in *Figure 1—figure supplement 1*.

**Figure supplement 2.** Career outcomes for EMBL alumni at five different time points for three different cohorts.

**Figure supplement 2—source data 1.** Summary data plotted in *Figure 1—figure supplement 2*.

**Figure supplement 3.** Comparing EMBL alumni with alumni from institutions in Canada and the United States.

We did not analyse the data for other career outcomes, as the smaller numbers of individuals in these careers made it difficult to identify real trends. Moreover, only a small number of institutions have released detailed data on the career destinations of recent postdoc alumni, and we are not aware of any long-term cohort-based data.

**Table 3.** Overview of PhD and postdoc cohorts.

| Completion years | PhD cohorts | | | Postdoc cohorts | | | All |
|---|---|---|---|---|---|---|---|
| | 1997–2004 | 2005–2012 | 2013–2020 | 1997–2004 | 2005–2012 | 2013–2020 | All |
| n = | 256 | 341 | 372 | 369 | 422 | 524 | 2284 |
| n (%) known current role | 225 (88%) | 306 (90%) | 332 (89%) | 336 (88%) | 364 (86%) | 472 (90%) | 2035 (89%) |
| n (%) detailed career path | 220 (70%) | 258 (79%) | 413 (77%) | 179 (60%) | 271 (61%) | 285 (79%) | 1626 (71%) |
| n (%) female | 85 (33%) | 157 (46%) | 173 (47%) | 136 (37%) | 149 (35%) | 207 (40%) | 907 (40%) |

### The proportion of EMBL alumni who become PIs has decreased with time

To estimate the probability of alumni from different cohorts entering a specific career each year after completing a PhD or postdoc at EMBL, we fitted the data to a Cox proportional hazards model. This is a statistical regression method that is commonly used to model time-to-event distributions from observational data with censoring (i.e., when not all study subjects are monitored until the event occurs, or the event never occurs for some of the subjects). In brief, we fitted the data to a univariate Cox proportional hazards model to calculate hazard ratios, which represent the relative chance of the event considered (here: entering a specific career) occurring in each cohort with respect to the oldest cohort. We also calculated Kaplan–Meier estimators, which estimate the probability of the event (entering a specific career) at different timepoints.

For both PhD and postdoc alumni entering PI positions, we observe hazard ratios of less than one in the Cox models when comparing the newer cohorts with the oldest cohort (Table S4 in *Supplementary file 1*), which indicates that the chances of becoming a PI have become lower for the newer cohorts. The Kaplan–Meier plots illustrate lower percentages of PIs among alumni from the most recent cohorts compared to the oldest cohort at equivalent timepoints (*Figure 2A*). Nevertheless, becoming a PI remained the most common career path for alumni from the 2005–2012 cohort (90/341=26.4% for PhD alumni) and (123/422=29.1% for postdoc alumni), and the most recent cohort of alumni appear to be on a similar trajectory.

Kaplan–Meier plots show increased proportions of the 2005–2012 and 2013–2020 cohorts entering Science-related Non-research and Non-science-related positions, compared to the 1997–2004 cohort for both PhD and postdoc alumni (*Figure 2D, E*). For the most recent (2013–2020) cohort, there was also an increased rate of entry into Industry: Research positions compared to alumni from PhD and postdoc cohorts from 1997 to 2004 and 2005–2012 (*Figure 2C*, Table S4 in *Supplementary file 1*). For Academia: other positions, the rate of entry was similar for all three PhD cohorts, though some differences between cohorts were observed for postdoc alumni (*Figure 2B*).

### A small increase in time between year of PhD and first PI position

We decided to explore to what extent increasing postdoc length may contribute to the decreased proportion of alumni who are found as PIs in the years after leaving EMBL. In order to fairly compare alumni from different cohorts, we included only alumni for whom we had a detailed career path, who had defended their PhD at least nine years ago, and who had become a PI within nine years of defending their PhD. We chose a nine-year cut-off because this was the time interval between the last PhDs in the 2005–2012 cohort and the execution of this study; moreover, for PhD alumni from the oldest cohort (1997–2004), most of those who became PIs had done so within nine years (89/97=92%).

157 of the PhD alumni in our sample met these criteria, taking an average of 5.6 calendar years to become a PI (see Methods). There was a statistically significant difference in the average time from PhD to first PI position between the 1997–2004 cohort (5.2 years) and the 2005–2012 cohort (6.1 years; *Figure 2—figure supplement 1A*). 218 of the postdoc alumni in our sample met these criteria, taking an average of 2.5 calendar years to become a PI after leaving EMBL (see Methods). There was no statistically significant difference in time between EMBL and first PI role for the 1997–2004 and 2005–2012 postdoc cohorts (*Figure 2—figure supplement 1B*). However, the time between receiving their PhD and becoming a PI increased by from 5.3 calendar years for the 1997–2004 postdoc cohort to 6.0 calendar years for the 2005–2012 postdoc cohort (*Figure 2—figure supplement 1C*).

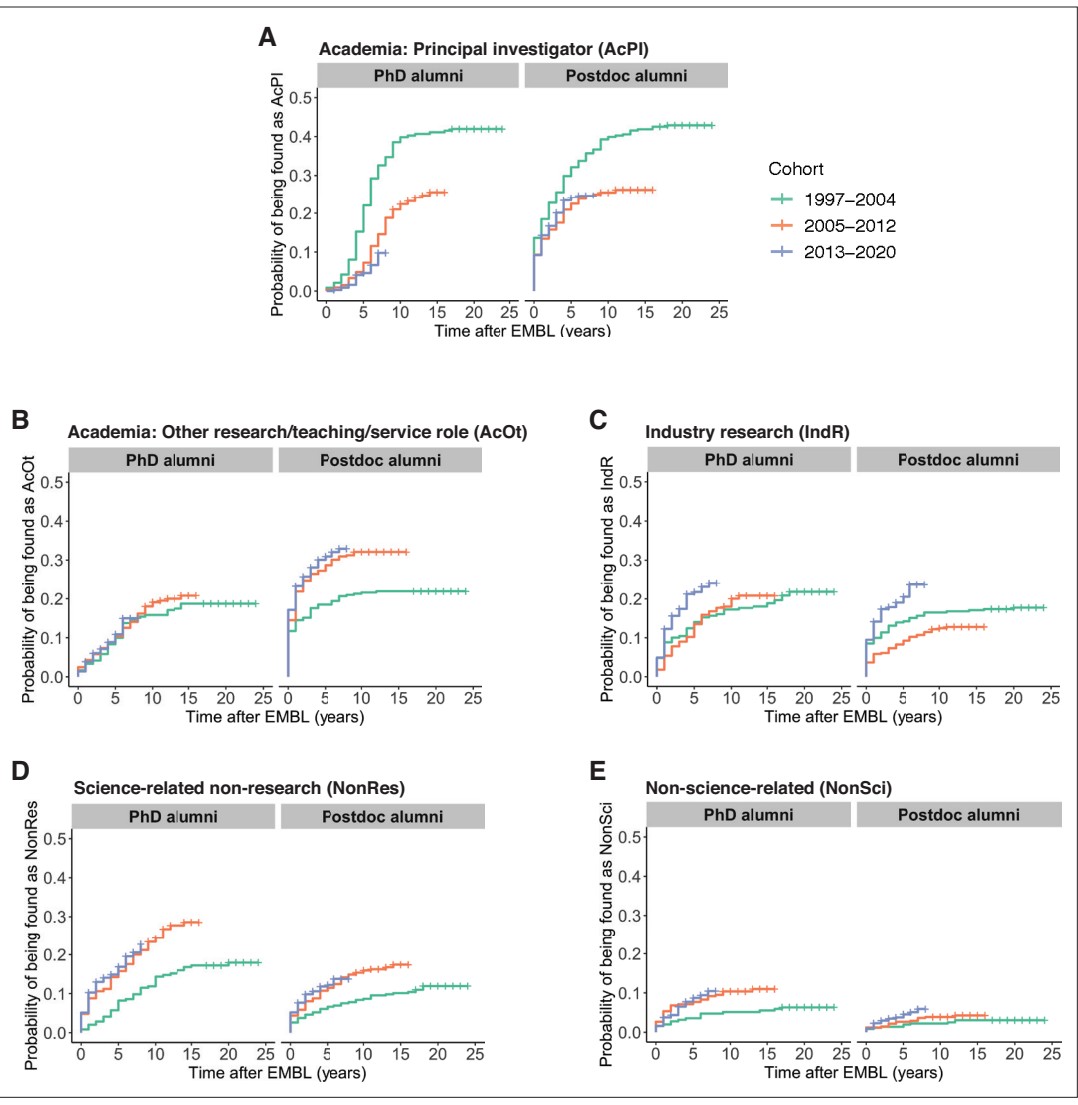

**Figure 2.** Changes in career outcomes for more recent cohorts. (**A**) Kaplan–Meier plots showing the estimated probability of an individual being in a PI position (y-axis) as a function of time after EMBL (x-axis) for three cohorts of PhD alumni (left) and three cohorts of postdoc alumni (right). Time after EMBL refers to the number of calendar years between PhD defence or leaving the EMBL postdoc programme and first PI position. (**B–E**) Similar Kaplan–Meier plots for Academia: Other positions (**B**), Industry Research positions (**C**), Science-related Non-research positions (**D**), and Non-science-related professions (**E**). Hazard ratios calculated by a Cox regression model can be found in Table S4 in *Supplementary file 1*.

The online version of this article includes the following source data and figure supplement(s) for figure 2:

**Figure supplement 1.** Length of time taken to become a PI.

**Figure supplement 1—source data 1.** Summary data plotted in *Figure 2—figure supplement 1*.

### Gender differences in career outcomes

Many studies have reported that female early-career researchers are less likely to remain in academia (*Alper, 1993*; *Martinez et al., 2007*). Consistent with these studies, male alumni from EMBL were more likely than female alumni to end up in a PI position (*Figure 3A and B*; Table S5 in *Supplementary file 1*). However, for alumni from 1997 to 2012, there was no statistically significant

difference in the length of time taken by male and female alumni to become PIs (*Figure 3—figure supplement 1*). Female alumni were more likely to end up in a Science-related Non-research position, and male alumni were more likely to end up in an Industry Research or Non-science-related position (*Figure 3*; *Figure 3—figure supplement 2*). However, female alumni were also more likely to be classified as unknown, and since it is more

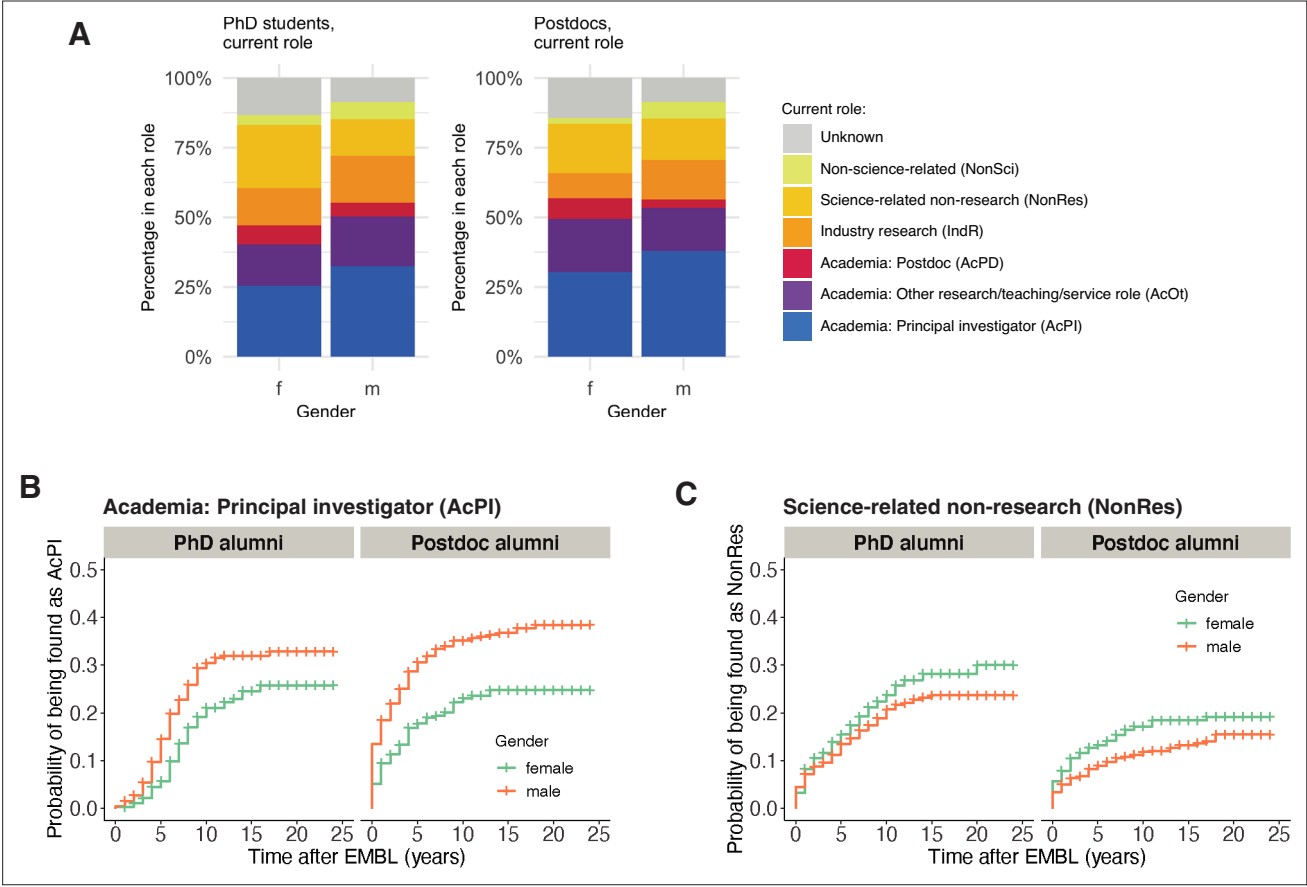

**Figure 3.** Gender differences in career outcomes. (**A**) Charts showing the percentage of female (n=415) and male (n=554) PhD alumni, and female (n=492) and male (n=823) postdoc alumni, in different careers in 2021. (**B**) Kaplan–Meier plots showing the estimated probability of an individual being in a PI position (y-axis) as a function of time after EMBL (x-axis), stratified by gender for PhD alumni (left) and postdoc alumni (right). (**C**) Kaplan–Meier plots showing the estimated probability of an individual being in a science-related non-research position as a function of time after EMBL, stratified by gender for PhD alumni (left) and postdoc alumni (right). Kaplan–Meier plots for other career outcomes are shown in *Figure 3—figure supplement 2*. Hazard ratios calculated by a Cox regression model can be found in Table S5 in *Supplementary file 1*.

The online version of this article includes the following source data and figure supplement(s) for figure 3:

**Source data 1.** Summary data plotted in *Figure 3A*.

**Figure supplement 1.** Length of time taken to become a PI for female and male alumni.

**Figure supplement 1—source data 1.** Summary data plotted in *Figure 3—figure supplement 1*.

**Figure supplement 2.** Rate of entry into different types of role by gender.

difficult to follow careers outside the academic world, it is possible that the number of women who established careers outside academia (in positions such as Industry Research, Science-related Non-research, and Non-science-related) is higher than our results suggest.

***Future PIs, on average, published more papers while at EMBL***
Publication metrics have been linked to the likelihood of obtaining (*van Dijk et al., 2014*; *Tregellas et al., 2018*) and succeeding (*von Bartheld et al., 2015*) in a faculty position. In this study, alumni who became PIs had more

favourable publication metrics from their EMBL work – for example, they published more articles, and their papers had higher CNCI values. (CNCI is short for Category Normalized Citation Impact, and a CNCI value of one means that the number of citations received was the same as the average for other articles in that field published in the same year; *Figure 4A and B*; Table S6 in *Supplementary file 1*). Using univariate Cox models for time to PI as a function of number of first-author research articles from EMBL work, we estimated that a postdoc with one first-author publication was 3.2 times more likely to be found in a PI position than a postdoc without a first-author

publication (95% confidence interval [2.2, 4.7]), and a post-doc with two or more first-author publications was 6.6 times more likely (95% confidence interval [4.7, 9.3]; *Figure 4C*).

### Publication factors are highly predictive of entry into a PI position

To understand the potential contribution of publication record in the context of other factors – including cohort, gender, nationality, publications, and seniority of the supervising PI – we fitted multivariate Cox models. To quantify publication record, we considered a range of metrics including journal impact factor, which has been shown to statistically correlate with becoming a PI in some studies (*van Dijk et al., 2014*) and has been used by some institutions in research evaluation (*McKiernan et al., 2019*). It should be stressed, however, that EMBL does not use journal impact factors in hiring or evaluation decisions, and is a signatory of the San Francisco Declaration on Research Assessment (DORA) and a member of the Coalition for Advancing Research Assessment (CoARA).

To evaluate the predictive power of each Cox model, we used the cross-validated Harrell's C-index, which measures predictive power as the average agreement across all pairs of individuals between observed and predicted temporal order of the outcome (in our case, entering a specific type of position; see Methods). A C-index of 1 indicates complete concordance between observed and predicted order. For example, for a model of entry into PI roles, a C-index of 1 would mean that the model correctly predicts, for all pairs of individuals, which individual becomes a PI first based on the factors included in the model. A C-index 0.5 is the baseline that corresponds to random guessing. Prediction is clearly limited by the fact that we could not explicitly encode some covariates that are certain to play an important role in career outcomes, such as career preferences and relevant skills. Nevertheless, the C-index for models containing all data were between 0.61 (entry to Industry Research, *Figure 4—figure supplement 1B*) and 0.70 (entry into PI positions, *Figure 4D*), suggesting that the factors have some predictive power.

To investigate which factors were most predictive for entry into different careers, we compared models containing different sets of factors. Consistent with previous studies, we found that statistics related to publications were highly predictive for entry into a PI position: a multivariate model containing only the publication statistics performs almost as well as the complete multivariate model, reaching a C-index of 0.69 (*Figure 4D*). The publications of the research group the alumnus was trained in (judged by the aggregated publication statistics for all PhD students and postdocs who were trained in the same group) was also predictive, with a C-index of 0.61.

Cohort/year, gender, and status at EMBL (PhD or postdoc) were also predictors of entry into a PI position in our Cox models, with C-indexes of 0.59, 0.57 and 0.55, respectively (*Figure 4D*). This is consistent with our observation that alumni from earlier cohorts/years (*Figure 1B*), male alumni (*Figure 3A*) and postdoc alumni (*Figure 1A*) were more frequently found in PI positions. Models containing only nationality or group leader seniority were not predictive.

For Academia: Other positions, the factors that were most predictive were those related to publications of the research group the alumnus was trained in (*Figure 4—figure supplement 1A*). It is unclear why this might be, but we speculate that this could reflect publication characteristics specific to certain fields that have a high number of staff positions, or other factors such as the scientific reputation, breadth or collaborative nature of the research group and its supervisor. The group's publications were also predictive for Industry Research and Science-related Non-research positions.

Time-related factors (i.e., cohort, PhD award year and EMBL contract start/end years) were the strongest prediction factors for Industry Research, Science-related Non-research, and Non-science-related positions (*Figure 4—figure supplement 1B–D*), and more recent alumni were more frequently found in these careers (*Figure 2C–E*).

Overall, statistics related to an individual's own publications were a weak predictor for entry into positions other than being a PI (*Figure 4—figure supplement 1*; *Figure 4—figure supplement 2*; Table S7–S11 in *Supplementary file 1*). For example, for Industry Research, a model containing statistics for an individual's publications had a C-index of only 0.53, compared to 0.61 for the complete model, and there were no differences in likelihood of a PhD alumnus with 0, 1 or 2+publications entering an Industry Research position.

### Changes in the publications landscape

Reports suggest that the number of authors on a typical research article in biology has increased over time, as has the amount of data in a typical article (*Vale, 2015*; *Fanelli and Larivière, 2016*);

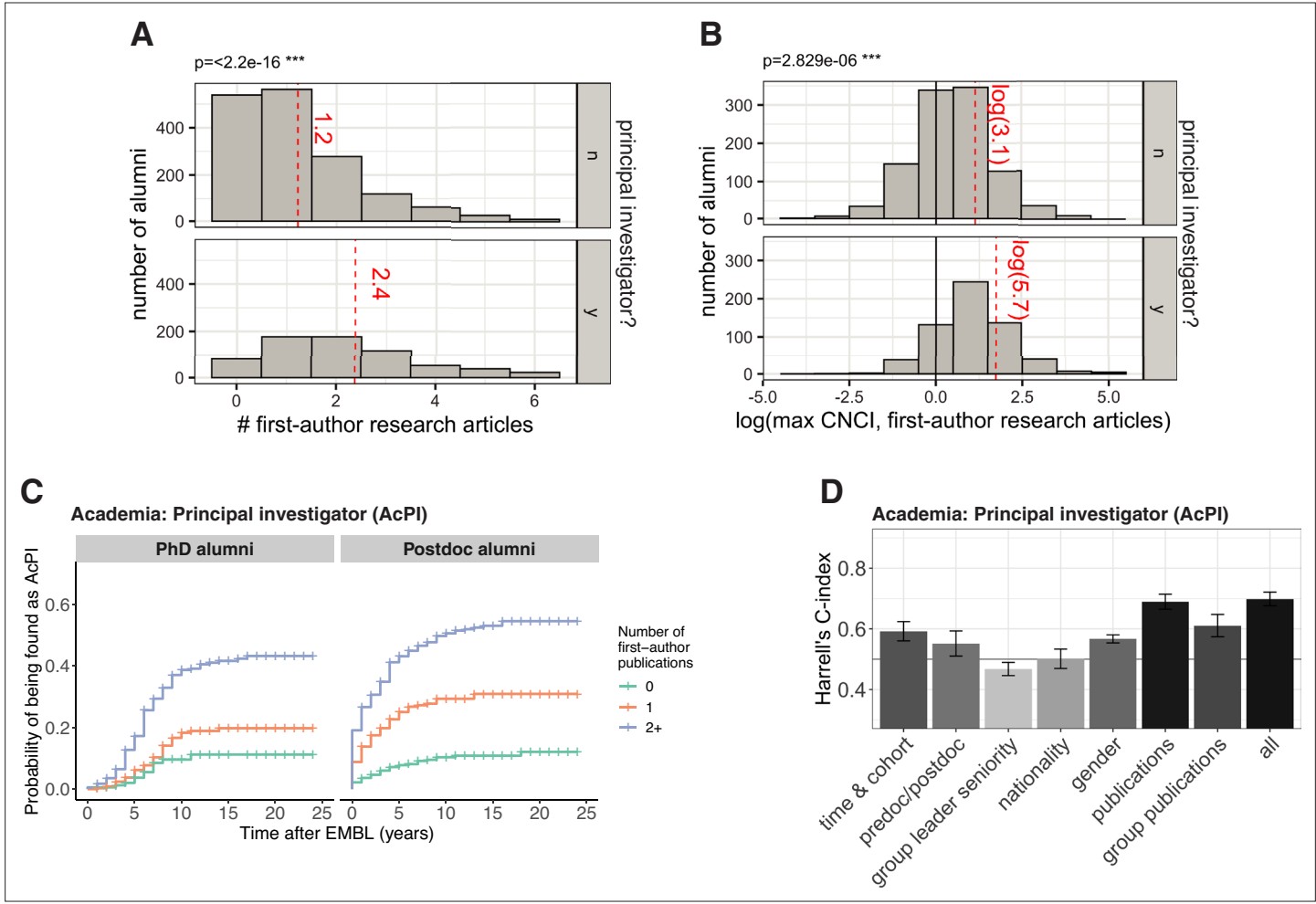

**Figure 4.** Publication factors are highly correlated with becoming a PI. (**A**) Histograms showing the number of alumni who have 0, 1, 2, 3,... first-author articles from their time at EMBL and became PIs (bottom; n=662, excluding 23 outliers), and did not become PIs (top; n=1594, excluding 5 outliers). For clearer visualization, and to protect the identity of alumni with outlying numbers of publications, the x-axis is truncated at the 97.5th percentile. The mean for each group (including outliers) is shown as a red dashed line; alumni who became PIs have an average of 2.4 first-author articles from their time at EMBL, whereas other alumni have an average of 1.2 articles; this difference is significant ($P < 2.2 \times 10^{-16}$; Welch's t-test). (**B**) 1656 alumni had one or more first-author articles from their time at EMBL that had a CNCI value in the InCites database. For each of these alumni, the natural logarithm of the highest CNCI value was calculated, and these histograms show the number of alumni for which this natural logarithm is between –4.5 and –3.5, between –3.5 and –2.5, and so on; the bottom histogram is for alumni who became PIs, and the top histogram is for other alumni. A CNCI value of 1 (plotted here at ln(1)=0; vertical black line) means that the number of citations received by the article is the same as the average for other articles in that field published in the same year. The mean for each group is shown as a red dashed line; alumni who became PIs have an average CNCI of 5.7, whereas other alumni have an average CNCI of 3.1; this difference is significant ($P < 2.829 \times 10^{-6}$; Welch's t-test). (**C**) Kaplan–Meier plots showing the estimated probability of an individual becoming a PI (y-axis) as a function of time after EMBL (x-axis), stratified by number of first-author publications from research completed at EMBL, for PhD alumni (left) and postdoc alumni (right). Hazard ratios calculated by a Cox regression model can be found in Table S7 in *Supplementary file 1*. (**D**) Harrell's C-Index for various Cox models for predicting entry into PI positions. The first seven bars show the C-index for univariate and multivariate models for a subset of covariates (which subset is shown below the x-axis), and the eighth bar is for a multivariate model that includes the covariates from all subsets. The subsets are time & cohort (multivariate, including the variables: cohort, PhD year (if known), start and end year at EMBL), predoc (ie PhD student)/postdoc (univariate), group leader seniority (univariate), nationality (univariate), gender (univariate), publications (multivariate: containing variables related to the alumni's publications from their EMBL work; these are variables with a name beginning with "pubs" in Table S1 in *Supplementary file 1*) and group publications (multivariate: containing variables related to the aggregated publication statistics for all PhD students and postdocs who were trained in the same group; these are variables with a name beginning with "group_pubs" in Table S1 in *Supplementary file 1*). A value of above 0.5 indicates that a model has predictive power, with a value of 1.0 indicating complete concordance between predicted and observed order to outcome (e.g. entry into a PI position). Bars denote the mean, and the error bars show the 95% confidence intervals. A value of above 0.5 indicates that a model has predictive power, with a value of 1.0 indicating complete concordance between predicted and observed order to outcome (e.g. entry into a PI position). Bars denote the mean, and the error bars show the 95% confidence intervals.

The online version of this article includes the following source data and figure supplement(s) for figure 4:

*Figure 4 continued on next page*

*Figure 4 continued*

**Source data 1.** Summary data plotted in *Figure 4A, B and D*.

**Figure supplement 1.** Cox models for predicting entry into various careers.

Harrell's C-Index for various Cox models for predicting entry into Academia: Other (**A**), Industry Research (**B**), Science-related Non-research (**C**), and Non-science-related careers (**D**). As in *Figure 4D*, the first seven bars show the C-index for univariate and multivariate models containing subsets of variables, the eighth bar is for a multivariate model containing all variables, and a value of above 0.5 indicates that a model has predictive power.

**Figure supplement 1—source data 1.** Summary data plotted in *Figure 4—figure supplement 1*.

**Figure supplement 2.** Entry into various careers and number of first-author publications.

a corresponding decrease in the number of first-author research articles per early-career researcher has also been reported (*Kendal et al., 2022*). For articles linked to the PhD students and postdocs in this study, the mean number of authors per article has more than doubled between 1995 and 2020 (*Figure 5A*). The mean number of articles per researcher did not change between the three cohorts studied (*Figure 5B*; the mean was 3.6 articles per researcher), but researchers from the second and third cohorts published fewer first-author articles than those from the first cohorts (*Figure 5C*). However, more recent articles had higher CNCI values (*Figure 5D*). The proportion of EMBL articles that included international collaborators also increased from 47% in 1995 to 79% in 2020.

## Discussion

Many early-career researchers are employed on fixed-term contracts funded by project-based grants, sometimes for a decade or more (*OECD, 2021*; *Acton et al., 2019*), and surveys suggest that early-career researchers are concerned about career progression (*Woolston, 2020*; *Woolston, 2019*). We hope PhD students and postdocs will be reassured to learn that the skills and knowledge they acquire during their training are useful in a range of careers both inside and outside acaemica.

Further changes to the career landscape in the life sciences are likely in future, not least as a result of the long-term impacts of the COVID-19 pandemic (*Bodin, 2020*). It is essential, therefore, that early-career researchers are provided with opportunities to reflect on their strengths, to understand the wide range of career options available to them, and to develop new skills.

The provision of effective support for PhD students and postdocs will require input from different stakeholders – including funders, employers, supervisors and policy makers – and the engagement of the early-career researchers themselves. At EMBL, a career service was launched in 2019 for all PhD students and postdocs, building

on a successful EC-funded pilot project that offered career support to 76 postdocs in the EMBL Interdisciplinary Postdoc Programme. The EMBL Fellows' Career Service now offers career webinars and a blog to the whole scientific community as well as additional tailored support for EMBL PhDs and postdocs including individual career guidance, workshops, resources and events. Funders and policymakers may also need to reassess the sustainably of academic career paths, and to review how funding is allocated between project-based grants and mechanisms that can support PI and non-PI positions with longer-term stability. These measures will will also support equality, diversity and inclusion in science, particularly if paired with research assessment practices that consider factors that can affect apparent research productivity such as career breaks, teaching and service activities.

Factors related to publication are highly predictive of entry into PI careers, and one challenge for an early-career researcher hoping to pursue such a career is to balance the number of articles they publish with the subjective quality of these articles. The trend towards fewer first-author articles per researcher likely reflects a global trend towards articles with more authors and a greater focus on collaborative and/or interdisciplinary approaches to research. Working on a project that involves multiple partners provides an early-career researcher with the opportunity to develop a range of skills, including teamwork, leadership and creativity. Such projects also allow researchers to tackle challenging biological questions from new angles to advance in their field of research, something viewed very positively by academic hiring committees (*Hsu et al., 2021*; *Clement et al., 2020*; *Fernandes et al., 2020*); however, multi-partner interdisciplinary projects can also take longer to complete. It is therefore important that early-career researchers and their supervisors discuss the potential impact and challenges of (prospective) projects, and what can be done to reduce any risks. For example, open science practices – including author credit statements, FAIR data, and pre-printing – can make

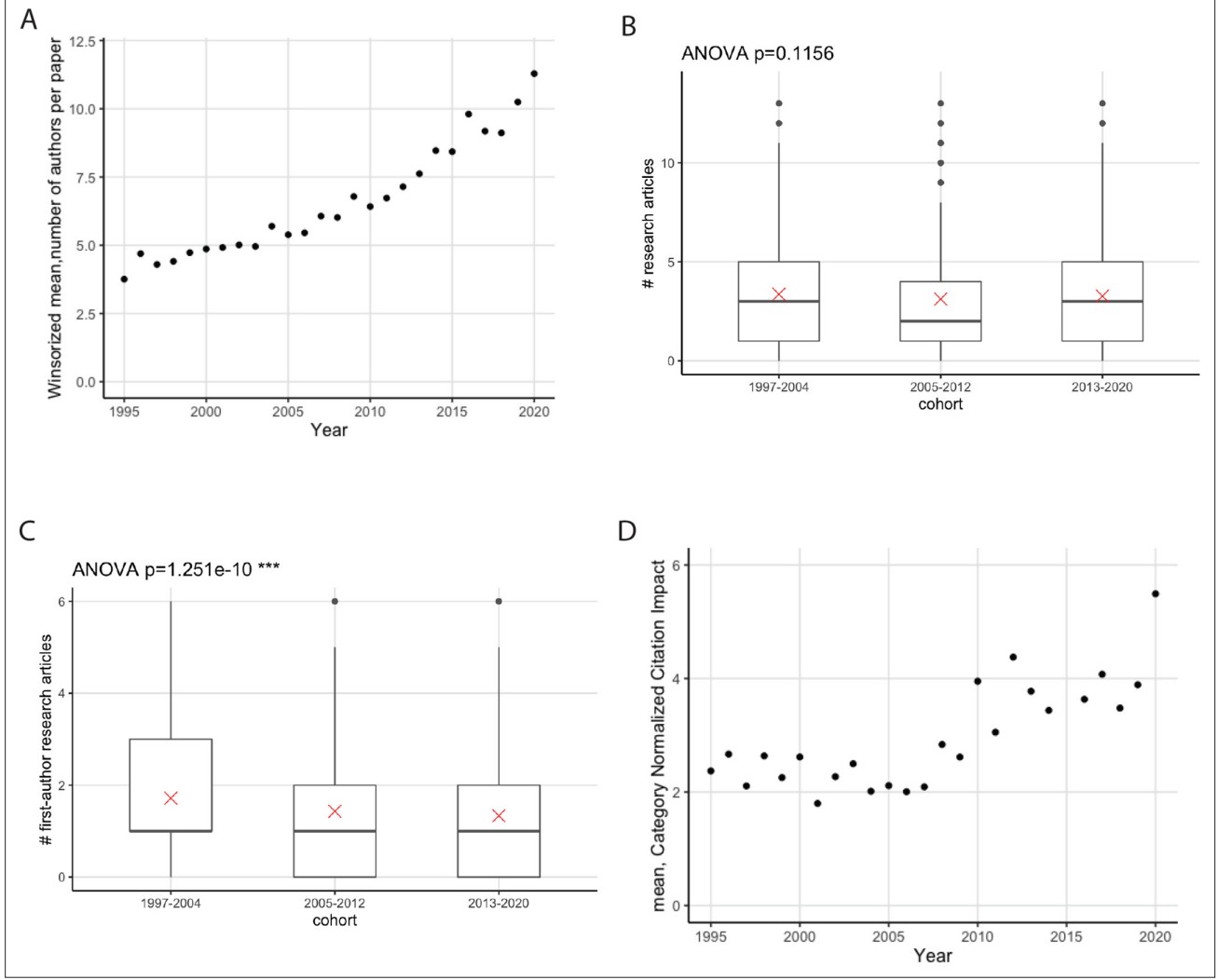

**Figure 5.** Publications are increasingly collaborative. (**A**) Mean number of authors (y-axis) as a function of year (x-axis) for research articles that were published between 1995 and 2020, and have at least one of the alumni included in this study as an author (n=5413); the winsorized mean has been used to limit the effect of outliers. The mean number of authors has increased by a factor of more than two between 1995 and 2000. (**B**) Boxplot showing the distribution of the number of articles published per researcher for three cohorts. The mean is indicated as a red cross; the circles are outliers. No statistically significant difference was found between the cohorts; the p-value of 0.1156 was generated using a one-way analysis of variance (ANOVA) test of the full dataset (including outliers); the p-value excluding outliers is 0.26. (**C**) Boxplot showing the distribution of the number of first-author articles published per researcher for three cohorts; the two most recent cohorts published fewer first-author articles than the 1997–2004 cohort; the p-value (excluding outliers) was $6.5 \times 10^{-7}$; see Table S12 in **Supplementary file 1**. (**D**) Mean CNCI (y-axis) as a function of time (x-axis) for research articles that were published between 1995 and 2020, and have at least one of the alumni included in this study as an author (n=5413). Recent articles have higher CNCI values. For clearer visualization, and to protect the identity of alumni with outlying numbers of publications, the y-axis in (**B**) and (**C**) is truncated at the 97.5 percentile.

The online version of this article includes the following source data for figure 5:

**Source data 1.** Summary data plotted in **Figure 5**.

project contributions more transparent and available faster (**Kaiser, 2017**; **McNutt et al., 2018**; **Wilkinson et al., 2016**; **Wolf et al., 2021**).

### Limitations
The limitations of our study include that its retrospective, observational design limits our ability to disentangle causation from correlation. The

changes in career outcomes may be driven primarily by increased competition for PI roles, but they could also be influenced by a greater availability or awareness of other career options. EMBL has held an annual career day highlighting career options outside academia since 2006, and many of our alumni decide to pursue a career in the private sector, attracted by perceptions of higher pay, more

stable contracts, and/or better work-life balance. Likewise, early-career researchers with an interest in a specific technology might, for example, prefer to work at a core facility.

Additionally, we cannot exclude the possibility that other factors may also affect the differences we see between cohorts (such as variations in the number of alumni taking up academic positions in countries that offer later scientific independence). Finally, although comparisons with data from the US and Canada suggest that the trend towards fewer alumni becoming PIs is a global phenomenon, it is possible that some of the trends we observe are specific to EMBL.

## Next steps

We plan to update our observational data every four years, and to maintain data on the career paths of alumni for 24 years after they leave EMBL. This will help us to identify any further changes in the career landscape and to better understand long-term career outcomes in the life sciences. *Silva et al., 2019* have also described a method for tracking career outcomes on a yearly basis with estimations of the time and other resources required. We encourage institutions to consider whether they can adapt our methods, or Silva's method, to the administrative processes and data-privacy regulations applicable at their institutions.

Future studies should also ideally include mixed-method longitudinal studies, which would allow information on career motivations, skills development, research environment, job application activity and other factors to be recorded. Combining the results of such studies with data on career outcomes would allow multifactorial and complex issues, such as gender differences in career outcomes, to be investigated, and would also provide policymakers with a fuller picture of workforce trends. Such studies would, however, require multiple institutions to commit to supplying large amounts of data every year, and coordinating the collection and analysis of such data year-on-year would be a major undertaking that would require the support of funders and institutions.

## Methods

### Data collection and analysis

The study includes individuals who graduated from the EMBL International PhD Programme between 1997 and 2020 (n=969), or who left the EMBL postdoc programme between 1997 and 2020 after spending at least one year as an EMBL postdoctoral fellow (n=1315). Each person is included only once in the study: where a PhD student remained at EMBL for a bridging or longer postdoc, they were included as PhD alumni only, with the postdoc position listed as a career outcome.

For each alumnus or alumna, we retrieved demographic information from our internal records and identified publicly available information about each person's career path (see *Supplementary file 2*). Where possible, this information was used to reconstruct a detailed career path. An individual was classified as having a "detailed career path" if an online CV or biosketch was found that accounted for their time since EMBL excluding a maximum of two one-calendar-year career breaks (which may, for example, reflect undisclosed sabbaticals or parental leave). Each position was classified using a detailed taxonomy, based on a published schema (*Stayart et al., 2020*), and given a broad overall classification (see *Supplementary file 2*). The country of the position was also recorded. For the most recent position, we noted whether the job title was indicative of a senior or management level role (i.e., if it included "VP"; "chief"; "cso";"cto"; "ceo"; "head"; "principal"; "president"; "manager"; "leader"; "senior"), or if they appeared to be running a scientific service or core facility in academia.

We use calendar years for all outcome data – for example, for an individual who left EMBL in 2012, the position one calendar year after EMBL would be the position held in 2013. If multiple positions were held in that year, we take the most recent position. We use calendar years, as the available online information often only provides the start and end year of a position (rather than exact date).

An EMBL publication record was also reconstituted for each person in the study. Each of their publications linked to EMBL in the Web of Science and InCites databases in June 2021 were recorded. The data included publication year and – for those indexed in InCites – crude metrics, such as CNCI, percentile in subject area, and journal impact factor. EMBL publications were assigned to individuals in the study based on matching name and publication year (see *Supplementary file 2* for full description). When an individual was the second author on a publication, we manually checked for declarations of co-first authorship. Aggregate publication statistics for individuals with the same primary supervisor were also calculated.

The names and other demographic information that would allow easy identification of individuals in the case of a data breach were pseudonymised. A file with key data for analysis and visualisation in R was then generated. A description of this data table can be found in Table S1 in *Supplementary file 1*, along with summary statistics.

## Statistical model

A Cox proportional hazards regression model was fitted to the data in order to predict time-to-event probabilities for each type of career outcome based on different covariates including cohort, publication variables and gender. Multivariate Cox models were fitted using a ridge penalty with penalty parameter chosen by 10-fold cross-validation. Harrell's C-index was calculated for each fit in an outer cross-validation scheme for validation and analysis of different models, with 10-fold cross-validation.

### Acknowledgements

We thank Monika Lachner and Anne Ephrussi for their critical reading of the manuscript and strong support of this project. We also acknowledge the instrumental support of the Alumni Relations, DPO, HR, SAP, Library, International PhD Programme and Postdoc Programme teams at EMBL. We also thank Edith Heard, Brenda Stride, Jana Watson-Kapps (FMI), and the Directorate, SAC, SSMAC and Council of the EMBL for discussion. The work was supported by: EMBL (JL, BK, MR, WH, RCG) and the EMBL International PhD Programme (BV). RCG is employed by EMBL's Interdisciplinary Postdoc Programme, which has received funding from the European Union's Horizon 2020 programme (Marie Skłodowska-Curie Actions).

**Junyan Lu**, Genome Biology Unit, European Molecular Biology Laboratory, Heidelberg, Germany
ⓘ http://orcid.org/0000-0002-9211-0746
**Britta Velten**, Genome Biology Unit, European Molecular Biology Laboratory, Heidelberg, Germany
ⓘ https://orcid.org/0000-0002-8397-3515
**Bernd Klaus**, Genome Biology Unit, European Molecular Biology Laboratory, Heidelberg, Germany
ⓘ https://orcid.org/0000-0003-1169-1225
**Mauricio Ramm**, EMBL International Centre for Advanced Training, European Molecular Biology Laboratory, Heidelberg, Germany
ⓘ https://orcid.org/0000-0001-8456-3246
**Wolfgang Huber**, Genome Biology Unit, European Molecular Biology Laboratory, Heidelberg, Germany
ⓘ https://orcid.org/0000-0002-0474-2218

**Rachel Coulthard-Graf**, EMBL International Centre for Advanced Training, European Molecular Biology Laboratory, Heidelberg, Germany
rachel.coulthard@embl.de
ⓘ https://orcid.org/0000-0003-1916-8498

*Author contributions:* Junyan Lu, Data curation, Formal analysis, Visualization, Methodology, Writing – review and editing; Britta Velten, Data curation, Formal analysis, Methodology, Visualization, Writing – review and editing; Bernd Klaus, Data curation, Formal analysis, Visualization, Methodology, Writing – review and editing; Mauricio Ramm, Investigation, Methodology; Wolfgang Huber, Supervision, Methodology, Writing – review and editing; Rachel Coulthard-Graf, Conceptualization, Data curation, Formal analysis, Investigation, Visualization, Methodology, Writing – original draft, Writing – review and editing

*Competing interests:* The authors declare that no competing interests exist.

### Funding

| Funder | Grant reference number | Author |
|---|---|---|
| Horizon 2020 Framework Programme | 664726 | Rachel Coulthard-Graf |
| Horizon 2020 Framework Programme | 847543 | Rachel Coulthard-Graf |
| European Molecular Biology Laboratory | | Britta Velten Bernd Klaus Mauricio Ramm Wolfgang Huber Rachel Coulthard-Graf Junyan Lu |

The funders had no role in study design, data collection and interpretation, or the decision to submit the work for publication.

### Decision letter and Author response

Decision letter https://doi.org/10.7554/eLife.78706.sa1
Author response https://doi.org/10.7554/eLife.78706.sa2

## Additional files

### Supplementary files

- MDAR checklist

- Supplementary file 1. Supplementary tables (Table S1–11).

- Supplementary file 2. Supplementary methods.

## Data availability

The data were collated for the provision of statistics, and are stored in a manner compliant with EMBL's internal policy on data protection. This policy means that the full dataset cannot be made publicly available (because the nature of the data means that sufficient anonymisation is not possible). Summary statistics for the main data table can be found in *Supplementary file 1* (Table S1). Rmarkdown documentation of the analysis and figures can be found here and is available on GitHub (copy archived at *Coulthard and Lu, 2022*).

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
