## [Decision Letter]

**Decision letter after peer review:**

Thank you for submitting your article "Meta-research: The changing career paths of PhDs and postdocs trained at EMBL" to *eLife* for consideration as a Feature Article. Your article has been reviewed by three peer reviewers, and the evaluation has been overseen by the *eLife* Features Editor (Peter Rodgers). The three reviewers have agreed to reveal their identity: Barbara Janssens; Sarvenaz Sarabipour; Reinhart Reithmeier.

The reviewers and editors have discussed the reviews and we have drafted this decision letter to help you prepare a revised submission. Please also note that *eLife* does not permit figures in supplementary materials: instead we allow primary figures to have figure supplements. I can suggest how to accommodate the figures in your supplementary materials if your article is accepted.

The *eLife* Features Editor will also contact you separately about some editorial issues that you will need to address.

Summary:

This paper deals with career outcomes for some 2284 PhD graduates and post-doctoral fellows from the European Molecular Biology Laboratories, a prestigious research institute that attracts top talent from around the world – the first of its kind. The paper is rich in data beyond simple career outcomes over time and includes gender, publications and collaboration. The methodology involved internet searches like other studies and was enhanced by robust statistical analyses. This important and timely study fills a gap in our knowledge, highlights the important role that institutions like EMBL play in training the next generation of researchers and innovators, and may stimulate other universities and research institutes to do the same. However, there are a number of points that need to be addressed to make the article suitable for publication.

Essential revisions:

1. It is a great achievement to show career destinations for 89% of the 2284 searched for. However the question is in which cohort the 249 "unknown" belong: it could be more transparent to keep those numbers included also in the detailed cohort analysis. It is also unclear, whether the cohort sizes reflect the actual number of researchers who left EMBL in that time period or whether data were lacking. The cohorts reported here increased about 33% from 2004 to 2012 and another 9% by 2020 (Table 2). Could it be, that for more recent cohorts more data are available – for example due to the fact that younger researchers can be found on social media like LinkedIn?

2. The main and interesting conclusion in the abstract is that of the 45% of alumni not continuing in academic research, one third does industry research and one third is in a science-related profession. Other interesting take-home messages are, that a large proportion of alumni changing sectors enter – or are quickly promoted to – managerial positions, that of those who entered a research position in industry one in ten returned to academia and that female Alumni were found less frequently in PI roles. It would be interesting to know, whether there is a difference in these numbers between cohorts – e.g. if more alumni return from industry to academia in recent cohorts and whether female researchers stay longer in postdoc roles (which could influence the total number of female PIs at a given time point).

3. In the time-resolved analysis the authors claim that the probability of being found as a PI in Academia diminished by about 15% after the first cohort (Figure 2C). However the question is whether the absolute number of PIs also decreased. This could be clarified with some reference numbers in the supplementary materials. When calculating the % of the given cohort sizes at different institutes (supp. Table 4) some differences (increase vs decrease) in the number of PIs can be found. Even if the hypothesis that the differences between cohorts are not EMBL-specific, and reflect a wide-spread change in the number of PhDs and postdocs relative to available PI positions seems valid, it would be good to clarify this issue.

4. Interesting findings is also the significant change in time between PhD and first PI position of 0.8 calendar years between the 2004 and 2012 cohorts. It is not surprising that publication factors are highly correlated with entry into PI positions: indeed all ECRs who became PIs published well, but, not all ECRs who published well became PIs! Publications have become more collaborative over the last decade (the number of coauthors has doubled and the number of first author publications per ECR diminished). Another relevant observation is the lack of correlation with group leader seniority or nationality. Group publications were also predictive for other research and science-related careers. Finally a strong observation is that 45% of leavers from the last 5-years who were found to be working outside of academia held senior or management-level roles. These findings can be reassuring for ECRs and the authors could consider to clearly state these in their conclusions.

5. Regarding the career tracking method used for this study: doing google searches for 2284 Alumni is a plausible effort and has probably been time consuming. A question for other research institutes and universities would be, whether this method of career tracking is scalable and/or feasible to continue as a regular task, or whether the authors see this as a one-time effort. If so, what kind or extent of career tracking would the authors recommend to continue sustainably? Performing career tracking is quite relevant, as institutes worldwide now start to be asked to deliver such data to governments and funding agents.

6. In Discussion and future work, it would be valuable to briefly discuss/aspire that institutions such as EMBL compile and publicly report on this type of data/records analysis together with surveys (what authors call mixed methods) of career intent and research environment.

i. With surveys one could have the number of EMBL trainees that actually applied for PI jobs.

ii. The fact that women were found less frequently in PI jobs does not reveal if (1) women apply less frequently or (2) search committees offer PI jobs less often to women or (3) combination of the two.

iii. Surveys together with the data presented in this work can examine the role of lab environment during training and job application.

7. An observation that authors have in the manuscript is that women are less represented as AcPIs (academic PIs). But it's not possible to claim it's an active mechanism. It would be valuable if authors plot the timeline by gender so readers could see the noise.

8. In various panels of Figures 2-4, please clarify if "Time after EMBL" (the label on the x-axis) means "Time after leaving EMBL" or "Time after arrival at EMBL".

Also, it appears that regardless of the cohort, postdocs have increased chances to become a PI only years after they leave EMBL, why? did they go on to do a second postdoc?

9. Please discuss supplementary table 4 in the text, and highlight any common findings from these studies.

10. The authors may also wish to comment on how some members of faculty recruitment committees may need to be trained to recognize bias in relying too heavily on citation indices and first author publications in hiring decisions rather than the scientific contribution of highly-qualified candidates to collaborative projects.

---

## [Author Response]

Essential revisions:1. It is a great achievement to show career destinations for 89% of the 2284 searched for. However the question is in which cohort the 249 "unknown" belong: it could be more transparent to keep those numbers included also in the detailed cohort analysis. It is also unclear, whether the cohort sizes reflect the actual number of researchers who left EMBL in that time period or whether data were lacking. The cohorts reported here increased about 33% from 2004 to 2012 and another 9% by 2020 (Table 2). Could it be, that for more recent cohorts more data are available – for example due to the fact that younger researchers can be found on social media like LinkedIn?

Thank you for these comments, we are happy to clarify here and in the manuscript:

The cohort sizes reflect the number of all PhD students and postdoctoral fellows who left in that time period, according to the official institutional administrative records. The organization grew during the time period included in the study (1997-2020), and the number of PhD students and postdocs in the later cohorts is therefore greater. It excludes individuals who were marked as deceased in our alumni records at the point the data was originally shared with us (June 2017 for 1997-2016 leavers and April 2021 for 2017-2020 leavers; or whose death we learned of during the update of our career tracking data in summer 2021 (through an updated alumni record, or if we found an online obituary)).

For each PhD or postdoc cohort, the percentage of alumni whose current position is unknown is between 9% and 14%, with no consistent trends with time. However, for the oldest cohort we less frequently found a complete online CV or biosketch that was detailed enough to confirm the type of position held for the full-time span since EMBL, particularly for postdoc alumni. Fewer of the older cohort therefore have a detailed CV/career path and there is a higher percentage of unknowns for specific timepoints after EMBL.

Changes to manuscript:

We have added an additional row with the number and percentage of alumni with detailed career paths for each cohort in Table 2 – new row = n(%) detailed career path

We have also clarified that the increased cohort size is due to growth in sentence that refers to this table: “More recent cohorts were also larger (Table 2), reflecting growth of the organization between 1997 and 2020.”

Column charts showing type of position by cohort (Figure 1B and Figure 1 —figure supplement 2 as 1B, but for other time-points) now include all alumni, not just those with a full career path available.

2. The main and interesting conclusion in the abstract is that of the 45% of alumni not continuing in academic research, one third does industry research and one third is in a science-related profession. Other interesting take-home messages are, that a large proportion of alumni changing sectors enter – or are quickly promoted to – managerial positions, that of those who entered a research position in industry one in ten returned to academia and that female Alumni were found less frequently in PI roles. It would be interesting to know, whether there is a difference in these numbers between cohorts – e.g. if more alumni return from industry to academia in recent cohorts and whether female researchers stay longer in postdoc roles (which could influence the total number of female PIs at a given time point).‘*if more alumni return from industry to academia in recent cohorts’*

This is difficult to assess due to the different career lengths and small numbers transitioning from one type of career to another. For example, of 415 alumni who we have a career path for and had at least one Industry role, just 22 returned to a faculty position. Twelve of these were from the oldest cohort (of 117 who held an industry role), compared to 7 (of 178) for the most recent cohort. Similarly, for PI to industry, 21 transitions were observed from the 539 career paths – 16 from the oldest cohort (from 231) and 1 (from 128) in the newest. Given that the propensity to transition may also change with career length, it is difficult to make comparisons or detect meaningful trends from these small numbers.

“whether female researchers stay longer in postdoc roles (which could influence the total number of female PIs at a given time point).”

We did not observe a statistically significant difference in length between PhD and becoming a PI, but agree that his is interesting and that it should be included in the manuscript.

To add this to the manuscript we made three changes: Additional figure supplement showing the average PhD to PI length for male vs female alumni [as previous figure comparing 2005-2012 and 1997-2004 cohorts, but now comparing female vs male alumni] (Figure 3 —figure supplement 1) – this suggests that male and female researchers spend similar times in postdoc roles as the differences are not statistically significant.

We have now included Kaplan Maier plots by gender, which also illustrate the entry into PI (and other) roles with time (as Figure 3B -C and Figure 3 —figure supplement 2 additional plots for AcOt, IndR, NonSci).

We also expand discussion of these data in the main text – see below with new detail italicised. (note: to allow more detailed discussion without requiring repetition of the information, we have moved this section after the sections on changes in career outcomes, where the Kaplan-Meier and PhD to PI length are first discussed).

Gender differences in career outcomes

Many studies have reported that female ECRs are less likely to remain in an academic career (44, 45). Consistent with these previous studies, we observed that male alumni were found more frequently in PI roles (Figure 3A-B; Table S5 in Supplementary information). *Figure 3A-B; Table S5 in Supplementary information. There was no statistically significant difference in the time to obtain a PI role between male and female alumni for alumni from 1997-2012, who became PIs within 9-years (Figure 3—figure supplement 1). The difference in career outcomes is therefore unlikely to be explained by different career dynamics for male and female alumni.*

Female alumni were more frequently found in science-related non-research roles than male alumni (Figure 3A). *In our Cox models, there was also a statistically significant difference between genders in entry into science-related non-research roles for postdoc alumni [p = 0.016] (Figure 3C; Table S5 in Supplementary information).*

We more frequently found male alumni in industry research and non-science-related roles than their female colleagues (Figure 3A; Figure 3—figure supplement 2B-C). However, a higher percentage of female than male alumni could not be located. *As academics are usually listed on institutional websites, often with a historical publication list that allows unambiguous identification, we expect that most alumni who were not located are employed in the non-academic sector. This means that, considering the higher percentage of female alumni with unknown career paths (where non-academic careers are likely over-represented), the true percentage of female alumni in industry research and non-science-related roles is likely higher, and possibly comparable with the percentage of male alumni in these roles.*

3. In the time-resolved analysis the authors claim that the probability of being found as a PI in Academia diminished by about 15% after the first cohort (Figure 2C). However the question is whether the absolute number of PIs also decreased. This could be clarified with some reference numbers in the supplementary materials. When calculating the % of the given cohort sizes at different institutes (supp. Table 4) some differences (increase vs decrease) in the number of PIs can be found. Even if the hypothesis that the differences between cohorts are not EMBL-specific, and reflect a wide-spread change in the number of PhDs and postdocs relative to available PI positions seems valid, it would be good to clarify this issue.

We have added information on the absolute number of PIs for each group in the supplementary table that collates the data published from other institutions and comparable EMBL data (in original manuscript table 4; now Table S3 in Supplementary file 1). We have also expanded discussion of this table the text in response to comment 9 (see comment 9 below) and include the absolute numbers.

Additional clarification

In the datasets, the number of PhDs trained per year has increased with time at all institutions. The cohort size – and how much this has increased for more recent cohorts however varies – for example, Stanford’s 2011-2015 cohort was just 18% larger than its 2000-2005 cohort (503 vs 426), whilst the University of Toronto’s 2012-2015 is 96% larger than its 2000-2003 (1234 vs 629). For EMBL, the PhD cohort sizes increased 45% from 256 for the 1997-2004 cohort, to 372 for 2013-2020. Therefore, the absolute number of PIs with time is difficult to compare between institutions. We therefore feel that the percentage of alumni entering different career options is the most pragmatic measure for comparing how career outcomes are changing with time. It can be viewed as the ‘chance’ of a ECR from a specific programme of entering that career area. If an institution continues to train the same absolute number of PIs per year, but trained many more scientists, it nevertheless saw a big difference the career outcomes of its alumni, with more alumni entering non-PI roles.

4. Interesting findings is also the significant change in time between PhD and first PI position of 0.8 calendar years between the 2004 and 2012 cohorts. It is not surprising that publication factors are highly correlated with entry into PI positions: indeed all ECRs who became PIs published well, but, not all ECRs who published well became PIs! Publications have become more collaborative over the last decade (the number of coauthors has doubled and the number of first author publications per ECR diminished). Another relevant observation is the lack of correlation with group leader seniority or nationality. Group publications were also predictive for other research and science-related careers. Finally a strong observation is that 45% of leavers from the last 5-years who were found to be working outside of academia held senior or management-level roles. These findings can be reassuring for ECRs and the authors could consider to clearly state these in their conclusions.

We have made the following changes to the manuscript to emphasise the positive aspects of our career findings (but balance the editorial comment that “If possible please shorten the first paragraph of the discussion and avoid any unnecessary repetition of material from earlier in the article.”).

This now reads:

“Many ECRs are employed on fixed-term contracts funded by project-based grants, sometimes for a decade or more (52, 53), and surveys suggest that ECRs are concerned about career progression (18-22). We hope ECRs will be reassured by the results of our time-resolved analysis that indicate that the skills and knowledge developed as an ECR can be applied in academia, industry and other sectors. Within academic research, service and teaching, we observed a marked reduction in the percentage of alumni entering PI roles; nevertheless, academic careers continue to be an important career destination. The percentage of alumni entering career areas outside academic research, service and teaching has increased, and our data suggest that ECRs’ skills are valued in these careers; 45% of leavers from the last 5-years who were found to be working outside of academic research and teaching held senior or management-level roles.”

5. Regarding the career tracking method used for this study: doing google searches for 2284 Alumni is a plausible effort and has probably been time consuming. A question for other research institutes and universities would be, whether this method of career tracking is scalable and/or feasible to continue as a regular task, or whether the authors see this as a one-time effort. If so, what kind or extent of career tracking would the authors recommend to continue sustainably? Performing career tracking is quite relevant, as institutes worldwide now start to be asked to deliver such data to governments and funding agents.

We have expanded the Discussion section ‘Future career studies’ to include a recommendation as follows:

“Evaluating the outcomes of training programmes, and making these data transparently available is a valuable exercise that can provide information to policymakers, transparency for ECRs, and help institutions provide effective career development support. We plan to update our observational data every four years, and maintain data on the career paths of alumni for 24 years after they leave EMBL. This will help us to identify any further changes in the career landscape and better understand long-term career outcomes. Silva et al. (2019) (73) have also described a method for completing career outcome tracking on a yearly basis with estimations of the time and other resources required. We encourage institutions to consider whether they can adapt our, or Silva’s method to the administrative processes and data-privacy regulations applicable at their institutions.”

6. In Discussion and future work, it would be valuable to briefly discuss/aspire that institutions such as EMBL compile and publicly report on this type of data/records analysis together with surveys (what authors call mixed methods) of career intent and research environment.i. With surveys one could have the number of EMBL trainees that actually applied for PI jobs.ii. The fact that women were found less frequently in PI jobs does not reveal if (1) women apply less frequently or (2) search committees offer PI jobs less often to women or (3) combination of the two.iii. Surveys together with the data presented in this work can examine the role of lab environment during training and job application.

Changes to manuscript: We have re-written the ‘Future career studies’ section of the discussion to incorporate these points. This now reads:

“Future studies should also ideally include mixed-method longitudinal studies. This would allow career motivations, skills development, research environment, job application activity, and other factors to be recorded by surveys during ECR training. Correlating these factors to career and training outcomes would allow investigation of multifactorial and complex issues such as gender differences in career outcomes, and provide policymakers with a fuller picture of workforce trends. Such studies will, however, require large sample sizes from multiple institutions and would need significant time investment and coordination over a long time period. The commitment and the support of funders and institutions would therefore be critical.”

7. An observation that authors have in the manuscript is that women are less represented as AcPIs (academic PIs). But it's not possible to claim it's an active mechanism. It would be valuable if authors plot the timeline by gender so readers could see the noise.

Changes to manuscript: We have generated Kaplan Maier plots by gender for entry into each career area, and include these in Figure 3 and Figure 3 —figure supplement 2 (see discussion of reviewer comment 2, above). The hazard ratios, 95% confidence intervals and p values are provided in a supplementary table in file 1 so that the confidence can be judged.

8. In various panels of Figures 2-4, please clarify if "Time after EMBL" (the label on the x-axis) means "Time after leaving EMBL" or "Time after arrival at EMBL".

We have added a clarification in the figure legend (“Time after EMBL refers to the number of calendar years between defence of an EMBL PhD and first PI role (for PhD alumni)) – or number of calendar years between leaving the EMBL postdoc programme and first PI role (for postdoc alumni)”, or clarified this in the figure labels, for each figure.

Also, it appears that regardless of the cohort, postdocs have increased chances to become a PI only years after they leave EMBL, why? did they go on to do a second postdoc?

We have expanded on this in the Results section ‘EMBL alumni contribute to research and innovation in academic and non-academic roles’, adding the following text:

“On average, PhD alumni became PIs 6.1 calendar years after their PhD defence. For postdoc alumni, the start year of the first PI role was on average 7.3 calendar years after their PhD and 2.5 years after completing their EMBL postdoc. Almost half of EMBL postdoc alumni who became PIs did so directly after completing their EMBL postdoc (168 of 343 alumni with a detailed career path available). Other postdoc alumni made the transition later, most frequently after one additional postdoc (71 alumni) or a single academic research / teaching / service position (56 alumni). Some alumni held multiple academic (40 alumni), or one or more non-academic positions (8 alumni) between their EMBL postdoc and first PI role.”

And to provide balance / to avoid focusing only on academic careers, we also include an additional sentence in the subsequent paragraph on non-academic areas:

“The average time between being awarded a PhD and the first industry research, science-related or non-science-related role was 5.0, 5.3 and 4.2 calendar years respectively.”

9. Please discuss supplementary table 4 in the text, and highlight any common findings from these studies.

We have expanded the discussion of this in the Results section ‘The percentage of EMBL alumni becoming PIs is similar to data released by North American institutions for both older and more recent cohorts’. [note that due to rearrangements, supplementary table 4 is now ‘Table S3 in Supplementary file 1’]. This now reads:

“A number of institutions have released cohort-based PhD outcomes data online or in publications (32, 44-49). Of these, a recent dataset from Stanford University offers the longest career tracking, reporting outcomes for PhD graduates from 20 graduation years (2000-2019) (44). In this dataset, Stanford University reported that 34% (145/426) of its 2000-2005 PhD alumni were in research-focussed faculty roles in 2018. This is comparable to the 37% (78/210) we observe for the EMBL alumni for the same time period. For 2011-2015 graduates, comparable percentages of Stanford (13%, 63/503) and EMBL (11%, 25/234) graduates were also in PI roles in 2018 (Figure 1C). Figure 1 – Supplement 3 plots data from five other datasets alongside equivalent data from EMBL (further details of each dataset are available in the Table S3 in Supplementary file 1). This includes a published dataset from the University of Toronto (32, 49), which reported that 31% (192/629) of its 2000-2003 life science division graduates and 25% (203/816) of its 2004-2007 life science graduates were in tenure stream roles in 2016. The equivalent EMBL data is comparable at 39% (52/132) and 28% (49/172) of graduates in PI roles respectively. Across all six datasets, EMBL and the other institutes generally have a similar proportion of alumni entering PI roles for comparable cohorts. This is consistent with our hypothesis that the differences between cohorts are not EMBL-specific, and reflect a wide-spread change in the number of PhDs and postdocs relative to available PI positions.”

10. The authors may also wish to comment on how some members of faculty recruitment committees may need to be trained to recognize bias in relying too heavily on citation indices and first author publications in hiring decisions rather than the scientific contribution of highly-qualified candidates to collaborative projects.

We have expanded this section of the discussion (third paragraph of ‘Addressing ECR career challenges’ in the discussion), also mentioning the trend to narrative CVs that has accelerated recently. This now reads:

“Research assessment and availability of funding play an important role in determining the career prospects of an academic. Therefore, it is also vital that factors that may lead to misperception of the productivity of ECRs, such as involvement in large-scale projects, career breaks, or time spent on teaching and service, are considered in research assessment during hiring, promotion and funding decisions. Initiatives such as the San Francisco Declaration on Research Assessment (DORA) and Coalition for Advancing Research Assessment (CoARA) have been advocating for an increased focus on good practice in evaluating research outputs, and many institutions and funders have reviewed their practices. Cancer Research UK, for example, now asks applicants to its grants to describe three to five research achievements, which can include non-publication outputs (64) and narrative CV formats that allow candidates to put their achievements in context are also becoming more common (61). The impact of the coronavirus pandemic on research productivity of researchers with caregiving responsibilities makes such actions imperative (65-67). We welcome this increased focus on qualitative assessment of scientific contribution, rather than reliance on publication metrics. To ensure successful implementation and to avoid unintended consequences (such as introducing new biases), it will be important for funders and institutions to provide appropriate guidance and/or training to evaluators and to carefully monitor implementation. Other initiatives that may help ECR involved in ambitious projects to demonstrate their contributions include more transparent author contribution information in publications (68, 69) and promotion of "FAIR" principles of data management (70). The increasing use of preprints (41, 71) is also likely to have a positive effect on the careers of ECRs involved in projects with longer time scales (72).”